# LATENT CONVOLUTIONAL MODELS

**ShahRukh Athar**\*  **Evgeny Burnaev**  **Victor Lempitsky**†
Skolkovo Institute of Science and Technology (Skoltech), Russia

## ABSTRACT

We present a new latent model of natural images that can be learned on large-scale datasets. The learning process provides a latent embedding for every image in the training dataset, as well as a deep convolutional network that maps the latent space to the image space. After training, the new model provides a strong and universal image prior for a variety of image restoration tasks such as large-hole inpainting, superresolution, and colorization. To model high-resolution natural images, our approach uses latent spaces of very high dimensionality (one to two orders of magnitude higher than previous latent image models). To tackle this high dimensionality, we use latent spaces with a special manifold structure (convolutional manifolds) parameterized by a ConvNet of a certain architecture. In the experiments, we compare the learned latent models with latent models learned by autoencoders, advanced variants of generative adversarial networks, and a strong baseline system using simpler parameterization of the latent space. Our model outperforms the competing approaches over a range of restoration tasks.

## 1 INTRODUCTION

Learning good image priors is one of the core problems of computer vision and machine learning. One promising approach to obtaining such priors is to learn a deep latent model, where the set of natural images is parameterized by a certain simple-structured set or probabilistic distribution, whereas the complexity of natural images is tackled by a deep ConvNet (often called a generator or a decoder) that maps from the latent space into the space of images. The best known examples are generative adversarial networks (GANs) (Goodfellow et al., 2014) and autoencoders (Goodfellow et al., 2016).

Given a good deep latent model, virtually any image restoration task can be solved by finding a latent representation that best corresponds to the image evidence (e.g. the known pixels of an occluded image or a low-resolution image). The attractiveness of such approach is in the universality of the learned image prior. Indeed, applying the model to a new restoration task can be performed by simply changing the likelihood objective. The same latent model can therefore be reused for multiple tasks, and the learning process needs not to know the image degradation process in advance. This is in contrast to task-specific approaches that usually train deep feed-forward ConvNets for individual tasks, and which have a limited ability to generalize across tasks (e.g. a feed-forward network trained for denoising cannot perform large-hole inpainting and vice versa).

At the moment, such image restoration approach based on latent models is limited to low-resolution images. E.g. (Yeh et al., 2017) showed how a latent model trained with GAN can be used to perform inpainting of tightly-cropped $64 \times 64$ face images. Below, we show that such models trained with GANs cannot generalize to higher resolution (eventhough GAN-based systems are now able to obtain high-quality samples at high resolutions (Karras et al., 2018)). We argue that it is the limited dimensionality of the latent space in GANs and other existing latent models that precludes them from spanning the space of high-resolution natural images.

To scale up latent modeling to high-resolution images, we consider latent models with tens of thousands of latent dimensions (as compared to few hundred latent dimensions in existing works). We show that training such latent models is possible using direct optimization (Bojanowski et al., 2018) and that it leads to good image priors that can be used across a broad variety of reconstruction tasks. In previous models, the latent space has a simple structure such as a sphere or a box in a Euclidean space, or a full Euclidean space with a Gaussian prior. Such choice, however, is not viable in our

---

\*Currently at Stony Brook University.
†Currently also with Samsung AI Center, Moscow.

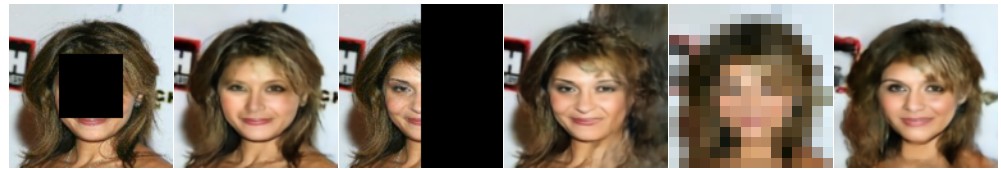

Figure 1: Restorations using the same Latent Convolutional Model (images 2,4,6) for different image degradations (images 1,3,5). At training time, our approach builds a latent model of non-degraded images, and at test time the restoration process simply finds a latent representation that maximizes the likelihood of the corrupted image and outputs a corresponding non-degraded image as a restoration result.

case, as vectors with tens of thousands of dimensions cannot be easily used as inputs to a generator. Therefore, we consider two alternative parameterizations of a latent space. Firstly, as a baseline, we consider latent spaces parameterized by image stacks (three-dimensional tensors), which allows to have "fully-convolutional" generators with reasonable number of parameters.

Our full system uses a more sophisticated parameterization of the latent space, which we call a *convolutional manifold*, where the elements of the manifold correspond to the parameter vector of a separate ConvNet. Such indirect parameterization of images and image stacks have recently been shown to impose a certain prior (Ulyanov et al., 2018), which is beneficial for restoration of natural images. In our case, we show that a similar prior can be used with success to parameterize high-dimensional latent spaces.

To sum up, our contributions are as follows. Firstly, we consider the training of deep latent image models with the latent dimensionality that is much higher than previous works, and demonstrate that the resulting models provide universal (w.r.t. restoration tasks) image priors. Secondly, we suggest and investigate the convolutional parameterization for the latent spaces of such models, and show the benefits of such parameterization.

Our experiments are performed on CelebA (Liu et al., 2015) (128x128 resolution), SUN Bedrooms (Yu et al., 2015) (256x256 resolution), CelebA-HQ (Karras et al., 2018) (1024x1024 resolution) datasets, and we demonstrate that the latent models, once trained, can be applied to large hole inpainting, superresolution of very small images, and colorization tasks, outperforming other latent models in our comparisons. To the best of our knowledge, we are the first to demonstrate how "direct" latent modeling of natural images without extra components can be used to solve image restoration problems at these resolutions (Figure 1).

**Other related work.**  Deep latent models follow a long line of works on latent image models that goes back at least to the eigenfaces approach (Sirovich & Kirby, 1987). In terms of restoration, a competing and more popular approach are feed-forward networks trained for specific restoration tasks, which have seen rapid progress recently. Our approach does not quite match the quality of e.g. (Iizuka et al., 2017), that is designed and trained specifically for the inpainting task, or the quality of e.g. (Yu & Porikli, 2016) that is designed and trained specifically for the face superresolution task. Yet the models trained within our approach (like other latent models) are universal, as they can handle degradations unanticipated at training time.

Our work is also related to pre-deep learning ("shallow") methods that learn priors on (potentially-overlapping) image patches using maximum likelihood-type objectives such as (Roth & Black, 2005; Karklin & Lewicki, 2009; Zoran & Weiss, 2011). The use of multiple layers in our method allows to capture much longer correlations. As a result, our method can be used successfully to handle restoration tasks that require exploiting these correlations, such as large-hole inpainting.

## 2  METHOD

Let $\{\mathbf{x}_1, \mathbf{x}_2, \dots, \mathbf{x}_N\}$ be a set of training images, that are considered to be samples from the distribution $X$ of images in the space $\mathcal{X}$ of images of a certain size that need to be modeled. In latent modeling, we introduce a different space $\mathcal{Z}$ and a certain distribution $Z$ in that space that is used to re-parameterize $\mathcal{X}$. In previous works, $\mathcal{Z}$ is usually chosen to be a Euclidean space with few dozen to few hundred dimensions, while our choice for $\mathcal{Z}$ is discussed further below.

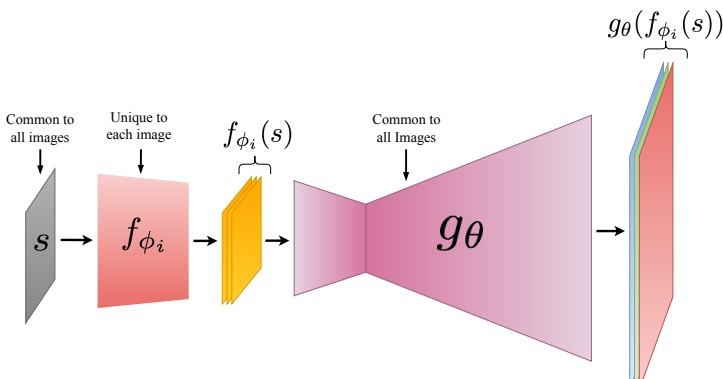

Figure 2: The Latent Convolutional Model incroprorates two sequential ConvNets. The smaller ConvNet $f$ (red) is fitted to each training image and is effectively used to parameterize the latent manifold. The bigger ConvNet $g$ (magenta) is used as a generator, and its parameters are fitted to all training data. The input $s$ to the pipeline is fixed to a random noise and not updated during training.

The deep latent modeling of images implies learning the generator network $g_\theta$ with learnable parameters $\theta$, which usually has convolutional architecture. The generator network maps from $\mathcal{Z}$ to $\mathcal{X}$ and in particular is trained so that $g_\theta(Z) \approx X$. Achieving the latter condition is extremely hard, and there are several approaches that can be used. Thus, generative adversarial networks (GANs) (Goodfellow et al., 2014) train the generator network in parallel with a separate discriminator network that in some variants of GANs serves as an approximate ratio estimator between $X$ and $X+g_\theta(Z)$ over points in $\mathcal{X}$. Alternatively, autoencoders (Goodfellow et al., 2016) and their variational counter-parts (Kingma & Welling, 2014) train the generator in parallel with the encoder operating in the reverse direction, resulting in a more complex distribution $Z$. Of these two approaches, only GANs are known to be capable of synthesizing high-resolution images, although such ability comes with additional tricks and modifications of the learning formulation (Arjovsky et al., 2017; Karras et al., 2018). In this work, we start with a simpler approach to deep latent modeling (Bojanowski et al., 2018) known as the GLO model. GLO model optimizes the parameters of the generator network in parallel with the explicit embeddings of the training examples $\{\mathbf{z}_1, \mathbf{z}_2, \ldots, \mathbf{z}_N\}$, such that $g_\theta(\mathbf{z}_i) \approx \mathbf{x}_i$ by the end of the optimization. Our approach differs from and expands (Bojanowski et al., 2018) in three ways: (i) we consider a much higher dimensionality of the latent space, (ii) we use an indirect parameterization of the latent space discussed further below, (iii) we demonstrate the applicability of the resulting model to a variety of image restoration tasks.

**Scaling up latent modeling.** Relatively low-dimensional latent models of natural images presented in previous works are capable of producing visually-compelling image samples from the distribution (Karras et al., 2018), but are not actually capable of matching or covering a rather high-dimensional distribution $X$. E.g. in our experiments, none of GAN models were capable of reconstructing most samples $\mathbf{x}$ from the hold-out set (or even from the training set; this observation is consistent with (Bojanowski et al., 2018) and also with (Zhu et al., 2016)). Being unable to reconstruct uncorrupted samples clearly suggests that the learned models are not suitable to perform restoration of corrupted samples. On the other hand, autoencoders and the related GLO latent model (Bojanowski et al., 2018) were able to achieve better reconstructions than GAN on the hold-out sets, yet have distinctly blurry reconstructions (even on the training set), suggesting strong underfitting.

We posit that existing deep latent models are limited by the dimensionality of the latent space that they consider, and aim to scale up this dimensionality significantly. Simply scaling up the latent dimensionality to few tens of dimensions is not easily feasible, as e.g. the generator network has to work with such a vector as an input, which would make the first fully-connected layer excessively large with hundreds of millions of parameters[1].

To achieve a tractable size of the generator, one can consider latent elements $\mathbf{z}$ to have a three-dimensional tensor structure, i.e. to be stacks of 2D image maps. Such choice of structure is very

---

[1]One can consider the first layer having a very thin matrix with a reasonable number of parameters mapping the latent vector to a much lower-dimensional space. This however would effectively amount to using lower-dimensional latent space and would defy the idea of scaling up latent dimensionality.

natural for convolutional architectures, and allows to train "fully-convolutional" generators with the first layer being a standard convolutional operation. The downside of this choice, as we shall see, is that it allows limited coordination between distant parts of the images $\mathbf{x} = g_\theta(\mathbf{z})$ produced by the generator. This drawback is avoided when the latent space is parameterized using latent convolutional manifolds as described next.

**Latent convolutional manifolds.** To impose more appropriate structure on the latent space, we consider structuring these spaces as *convolutional manifolds* defined as follows. Let $\mathbf{s}$ be a stack of maps of the size $W_s \times H_s \times C_s$ and let $\{f_\phi \,|\, \phi \in \Phi\}$ be a set of convolutional networks all sharing the same architecture $f$ that transforms $\mathbf{s}$ to different maps of size $W_z \times H_z \times C_z$. A certain parameter vector $\phi \in \Phi$ thus defines a certain convolutional network $f_\phi$. Then, let $\mathbf{z}(\phi) = f_\phi(\mathbf{s})$ be an element in the space of $(W_z \times H_z \times C_z)$-dimensional maps. Various choices of $\phi$ then span a manifold embedded into this space, and we refer to it as the *convolutional manifold*. A convolutional manifold $\mathbf{C}_{f,\mathbf{s}}$ is thus defined by the ConvNet architecture $f$ as well as by the choice of the input $\mathbf{s}$ (which in our experiments is always chosen to be filled with uniform random noise). Additionally, we also restrict the elements of vectors $\phi$ to lie within the $[-B; B]$ range. Formally, the convolutional manifold is defined as the following set:

$$\mathbf{C}_{f,\mathbf{s}} = \{\mathbf{z} \,|\, \mathbf{z} = f_\phi(\mathbf{s}), \phi \in \Phi\}, \ \ \Phi = [-B; B]^{N_\phi}, \tag{1}$$

where $\phi$ serves as a natural parameterization and $N_\phi$ is the number of network parameters. Below, we refer to $f$ as *latent ConvNet*, to disambiguate it from the generator $g$, which also has a convolutional structure.

The idea of the convolutional manifold is inspired by the recent work on deep image priors (Ulyanov et al., 2018). While they effectively use convolutional manifolds to model natural images directly, in our case, we use them to model the latent space of the generator networks resulting in a fully-fledged learnable latent image model (whereas the model in (Ulyanov et al., 2018) cannot be learned on a dataset of images). The work (Ulyanov et al., 2018) demonstrates that the regularization imposed by the structure of a very high-dimensional convolutional manifold is beneficial when modeling natural images. Our intuition here is that similar regularization would be beneficial in regularizing learning of high-dimensional latent spaces. As our experiments below reveal, this intuition holds true.

**Learning formulation.** Learning the deep latent model (Figure 2) in our framework then amounts to the following optimization task. Given the training examples $\{\mathbf{x}_1, \mathbf{x}_2, \ldots, \mathbf{x}_N\}$, the architecture $f$ of the convolutional manifold, and the architecture $g$ of the generator network, we seek the set of the latent ConvNet parameter vectors $\{\phi_1, \phi_2, \ldots, \phi_N\}$ and the parameters of the generator network $\theta$ that minimize the following objective:

$$L(\phi_1, \phi_2, \ldots, \phi_N, \theta) = \frac{1}{N} \sum_{i=1}^{N} \| g_\theta(f_{\phi_i}(\mathbf{s})) - \mathbf{x}_i \|, \tag{2}$$

with an additional box constraints $\phi_i^j \in [-0.01; 0.01]$ and $\mathbf{s}$ being a random set of image maps filled with uniform noise. Following (Bojanowski et al., 2018), the norm in (2) is taken to be the Laplacian-L1: $\|\mathbf{x}_1 - \mathbf{x}_2\|_{\text{Lap-L1}} = \sum_j 2^{-2j} |L^j(\mathbf{x}_1 - \mathbf{x}_2)|_1$, where $L^j$ is the $j$th level of the Laplacian image pyramid (Burt & Adelson, 1983). We have also found that adding an extra MSE loss term to the Lap-L1 loss term with the weight of 1.0 speeds up convergence of the models without affecting the results by much.

The optimization (2) is performed using stochastic gradient descent. As an outcome of the optimization, each training example $\mathbf{x}_i$ gets a representation $\mathbf{z}_i = f_{\phi_i}$ on the convolutional manifold $\mathbf{C}_{f,\mathbf{s}}$.

Importantly, the elements of the convolutional manifold then define a set of images in the image space (which is the image of the convolutional manifold under learned generator):

$$\mathbf{I}_{f,\mathbf{s},\theta} = \{\mathbf{x} \,|\, \mathbf{x} = g_\theta(f_\phi(\mathbf{s})), \phi \in \Phi\}. \tag{3}$$

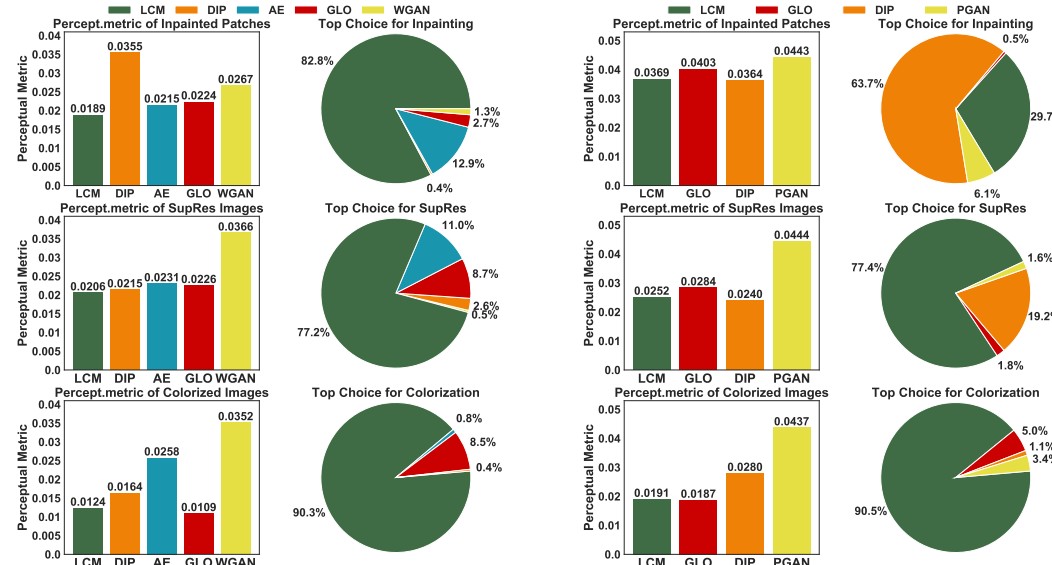

Figure 3: Results (perceptual metrics – lower is better – and user preferences) for the two datasets (CelebA – left, Bedrooms – right) and three tasks (inpainting, super-resolution, colorization). For the colorization task the perceptual metric is inadequate as the grayscale image has the lowest error, but is shown for completeness.

Table 1: MSE loss on the restored images with respect to the ground truth. For inpainting the MSE was calculated just over the inpainted region of the images.

| | CelebA | | | | | LSUN-Bedrooms | | | |
|---|---|---|---|---|---|---|---|---|---|
| | **LCM** | **GLO** | **DIP** | **AE** | **WGAN** | **LCM** | **GLO** | **DIP** | **PGAN** |
| **Inpainting** | 0.0034 | 0.0038 | 0.0091 | 0.0065 | 0.0344 | 0.0065 | 0.0085 | 0.0063 | 0.0097 |
| **Super-res** | 0.0061 | 0.0063 | 0.0052 | 0.0083 | 0.0446 | 0.0071 | 0.0069 | 0.0057 | 0.0183 |
| **Colorization** | 0.0071 | 0.0069 | 0.0136 | 0.0194 | 0.0373 | 0.0066 | 0.0075 | 0.0696 | 0.0205 |

While not all elements of the manifold $\mathbf{I}_{f,\mathbf{s},\theta}$ will correspond to natural images from the distribution $X$, we have found out that with few thousand dimensions, the resulting manifolds can cover the support of $X$ rather well. I.e. each sample from the image distribution can be approximated by the element of $\mathbf{I}_{f,\mathbf{s},\theta}$ with a low approximation error. This property can be used to perform all kinds of image restoration tasks.

**Image restoration using learned latent models.** We now describe how the learned latent model can be used to perform the restoration of the unknown image $\mathbf{x}_0$ from the distribution $X$, given some evidence $\mathbf{y}$. Depending on the degradation process, the evidence $\mathbf{y}$ can be an image $\mathbf{x}_0$ with masked values (inpainting task), the low-resolution version of $\mathbf{x}_0$ (superresolution task), the grayscale version of $\mathbf{x}_0$ (colorization task), the noisy version of $\mathbf{x}_0$ (denoising task), a certain statistics of $\mathbf{x}_0$ computed e.g. using a deep network (feature inversion task), etc.

We further assume, that the degradation process is described by the objective $E(\mathbf{x}|\mathbf{y})$, which can be set to minus log-likelihood $E(\mathbf{x}|\mathbf{y}) = -\log p(\mathbf{y}|\mathbf{x})$ of observing $\mathbf{y}$ as a result of the degradation of $\mathbf{x}$. E.g. for the *inpainting* task, one can use $E(\mathbf{x}|\mathbf{y}) = \|(\mathbf{x} - \mathbf{y}) \odot \mathbf{m}\|$, where $\mathbf{m}$ is the 0-1 mask of known pixels and $\odot$ denotes element-wise product. For the *superresolution* task, the restoration objective is naturally defined as $E(\mathbf{x}|\mathbf{y}) = \| \downarrow (\mathbf{x}) - \mathbf{y}\|$, where $\downarrow (\cdot)$ is an image downsampling operator (we use Lanczos in the experiments) and $\mathbf{y}$ is the low-resolution version of the image. For the *colorization* task, the objective is defined as $E(\mathbf{x}|\mathbf{y}) = \|\text{gray}(\mathbf{x}) - \mathbf{y}\|$, where $\text{gray}(\cdot)$ denotes a projection from the RGB to grayscale images (we use a simple averaging of the three color channels in the experiments) and $\mathbf{y}$ is the grayscale version of the image.

| Distorted Image | LCM (Ours) | GLO | DIP | WGAN | AE | Original Image |
|---|---|---|---|---|---|---|

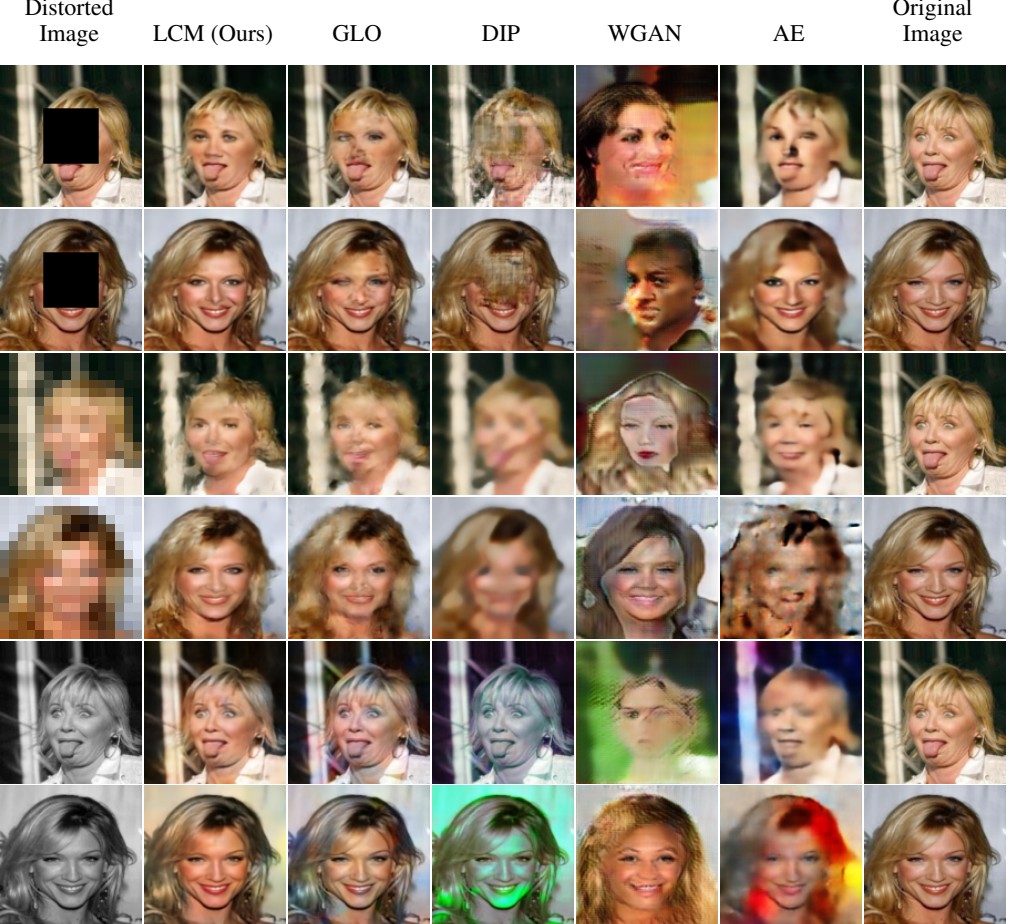

Figure 4: Qualitative comparison on CelebA (see the text for discussion).

Using the learned latent model as a prior, the following estimation combining the learned prior and the provided image evidence is performed:

$$\hat{\phi} = \arg\min_{\phi} E(g_\theta(f_\phi(\mathbf{s})) \,|\, \mathbf{y}), \qquad \hat{\mathbf{x}} = g_\theta(f_{\hat{\phi}}(\mathbf{s})). \qquad (4)$$

In other words, we simply estimate the element of the image manifold (3) that has the highest likelihood. The optimization is performed using stochastic gradient descent over the parameters $\phi$ on the latent convolutional manifold.

For the baseline models, which use a direct parameterization of the latent space, we perform analogous estimation using optimization in the latent space:

$$\hat{\mathbf{z}} = \arg\min_{\mathbf{z}} E(g_\theta(\mathbf{z}) \,|\, \mathbf{y}), \qquad \hat{\mathbf{x}} = g_\theta(\mathbf{z}). \qquad (5)$$

In the experiments, we compare the performance of our full model and several baseline models over a range of the restoration tasks using formulations (4) and (5).

## 3 EXPERIMENTS

**Datasets.** The experiments were conducted on three datasets. The **CelebA** dataset was obtained by taking the 150K images from (Liu et al., 2015) (cropped version) and resizing them from $178 \times 218$ to $128 \times 128$. Note that unlike most other works, we have performed anisotropic rescaling rather than additional cropping, leading to the version of the dataset with larger background portions and higher variability (corresponding to a harder modeling task). The **Bedrooms** dataset from the LSUN (Yu et al., 2015) is another popular dataset of images. We rescale all images to the $256 \times 256$ size. Finally, the **CelebA-HQ** dataset from (Karras et al., 2018) that consists of 30K $1024 \times 1024$ images of faces.

| Distorted Image | LCM (Ours) | GLO | DIP | PGAN | Original Image |

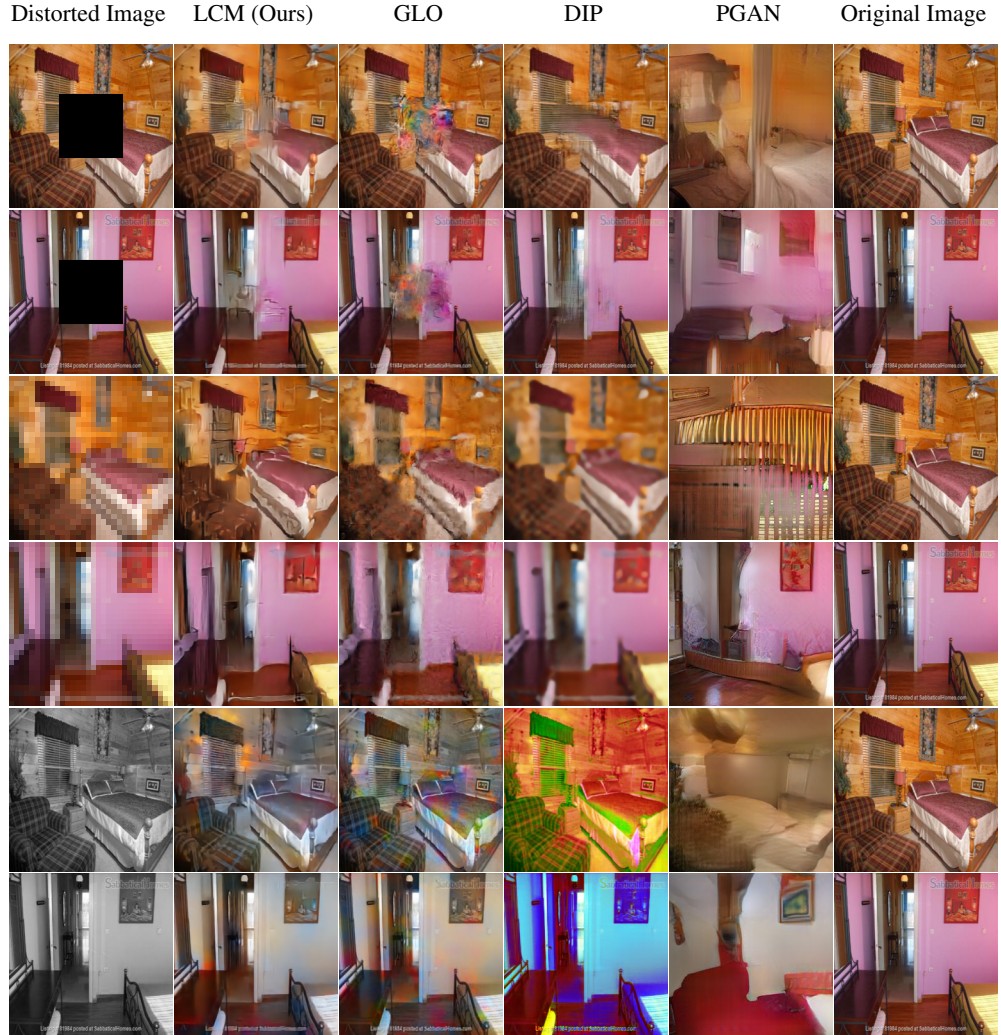

Figure 5: Qualitative comparison on SUN Bedrooms for the tasks of inpainting (rows 1-2), superresolution (rows 3-4), colorization (rows 5-6). The LCM method performs better than most methods for the first two tasks.

**Tasks.** We have compared methods for three diverse tasks. For the inpainting task, we have degraded the input images by masking the center part of the image ($50 \times 50$ for CelebA, $100 \times 100$ for Bedrooms, $400 \times 400$ for CelebA-HQ). For the superresolution task, we downsampled the images by a factor of eight. For the colorization task, we have averaged the color channels obtaining the gray version of the image.

## 3.1 EXPERIMENTS ON CELEBA AND BEDROOMS

We have performed extensive comparisons with other latent models on the two datasets with smaller image size and lower training times (CelebA and Bedrooms). The following latent models were compared:

- **Latent Convolutional Networks (LCM – Ours):** Each $f_{\phi_i}$ has 4 layers (in CelebA), 5 layers (in Bedrooms) or 7 layers (in CelebA-HQ) and takes as input random uniform noise. The Generator, $g_\theta$ has an hourglass architecture. The latent dimensionality of the model was 24k for CelebA and 61k for Bedrooms.
- **GLO**: The baseline model discussed in the end of Section 2 and inspired by (Bojanowski et al., 2018), where the generator network has the same architecture as in LCM, but the

convolutional space is parameterized by a set of maps. The latent dimensionality is the same as in LCM (and thus much higher than in (Bojanowski et al., 2018)). We have also tried a variant reproduced exactly from (Bojanowski et al., 2018) with vectorial latent spaces that feed into a fully-connected layers (for the dimensionalities ranging from 2048 to 8162 – see Appendix B), but invariably observed underfitting. Generally, we took extra care to find the optimal parameterization that would be most favourable to this baseline.

- **DIP**: The deep image prior-based restoration (Ulyanov et al., 2018). We use the architecture proposed by the authors in the paper. DIP can be regarded as an extreme version of our paper with the generator network being an identity. DIP fits 1M parameters to each image for inpainting and colorization and 2M parameters for super-resolution.

- **GAN**: For CelebA we train a WGAN-GP (Gulrajani et al., 2017) with the DCGAN type generator and a latent space of 256. For Bedrooms we use the pretrained Progressive GAN (PGAN) models with the latent space of dimensionality 512 published by the authors of (Karras et al., 2018). During restoration, we do not impose prior on the norm of $\mathbf{z}$ since it worsens the underfitting problem of GANs (as demonstrated in Appendix C).

- **AE**: For the CelebA we have also included a standard autoencoder using the Lap-L1 and MSE reconstruction metrics into the comparison (latent dimensionality 1024). We have also tried the variant with convolutional higher-dimensional latent space, but have observed very strong overfitting. The variational variant (latent dimensionality 1024) lead to stronger underfitting than the non-variational variant. As the experiments on CelebA clearly showed a strong underfitting, we have not included AE into the comparison on the higher-resolution Bedrooms dataset.

For Bedrooms dataset we restricted training to the first 200K training samples, except for the DIP (which does not require training) and GAN (we used the progressive GAN model trained on all 3M samples). All comparisons were performed on hold-out sets not used for training. Following (Bojanowski et al., 2018), we use plain SGD with very high learning rate of 1.0 to train LCM and of 10.0 to train the GLO models. The exact architectures are given in Appendix D.

**Metrics.** We have used quantitative and user study-based assessment of the results. For the quantitative measure, we have chosen the mean squared error (MSE) measure in pixel space, as well as the mean squared distance of the VGG16-features (Simonyan & Zisserman, 2015) between the original and the reconstructed images. Such *perceptual metrics* are known to be correlated with the human judgement (Johnson et al., 2016; Zhang et al., 2018). We have used the [relu1_2, relu2_2, relu3_3, relu4_3, relu5_3] layers contributing to the distance metric with equal weight. Generally, we observed that the relative performance of the methods were very similar for the MSE measure, for the individual VGG layers, and for the averaged VGG metrics that we report here. When computing the loss for the inpainting task we only considered the positions corresponding to the masked part.

Quantitative metrics however have limited relevance for the tasks with big multimodal conditional distributions, i.e. where two very different answers can be equally plausible, such as all three tasks that we consider (e.g. there could be very different colorizations of the same bedroom image).

In this situation, human judgement of quality is perhaps the best measure of the algorithm performance. To obtain such judgements, we have performed a user study, where we have picked 10 random images for each of the two datasets and each of the three tasks. The results of all compared methods alongside the degraded inputs were shown to the participants (100 for CelebA, 38 for Bedrooms). For each example, each subject was asked to pick the best restoration variant (we asked to take into account both realism and fidelity to the input). The results were presented in random order (shuffled independently for each example). We then just report the percentage of user choices for each method for a given task on a given dataset averaged over all subjects and all ten images.

**Results.** The results of the comparison are summarized in Figure 3 and Table 1 with representative examples shown in Figure 4 and Figure 5. "Traditional" latent models (built WGAN/PGAN and AE) performed poorly. In particular, GAN-based models produced results that were both unrealistic and poorly fit the likelihood. Note that during fitting we have not imposed the Gaussian prior on the latent space of GANs. Adding such prior did not result in considerable increase of realism and lead to even poorer fit to the evidence (see Appendix C).

The DIP model did very well for inpainting and superresolution of relatively unstructured Bedrooms dataset. It however performed very poorly on CelebA due to its inability to learn face structure from data and on the colorization task due to its inability to learn about natural image colors.

Except for the Bedrooms-inpainting, the new models with very large latent space produced results that were clearly favoured by the users. LCM performed better than GLO in all six user comparisons, while in terms of the perceptual metric the performance of LCM was also better than GLO for inpainting and superresolution tasks. For the colorization task, the LCM is unequivocally better in terms of user preferences, and worse in terms of the perceptual metric. We note that, however, perceptual metric is inadequate for the colorization task as the original grayscale image scores better than the results of all evaluated methods. We therefore only provide the results in this metric for colorization for the sake of completeness (finding good quantitative measure for the highly-ambiguous colorization task is a well-known unsolved problem).

Additional results on CelebA and Bedrooms dataset are given in Appendices A, F, G.

Table 2: Metrics of optimization over the z-space, the convolutional manifold and Progressive GAN (Karras et al., 2018) latent space

| Optimization Over | MSE (known pixels) | MSE (inpainted pixels) | Perceptual Metric |
|---|---|---|---|
| Convolutional Net Parameters | 0.00307 | 0.00171 | 0.02381 |
| Z-Space | 0.00141 | 0.00854 | 0.02736 |
| PGAN Latent Space | 0.00477 | 0.00224 | 0.02546 |

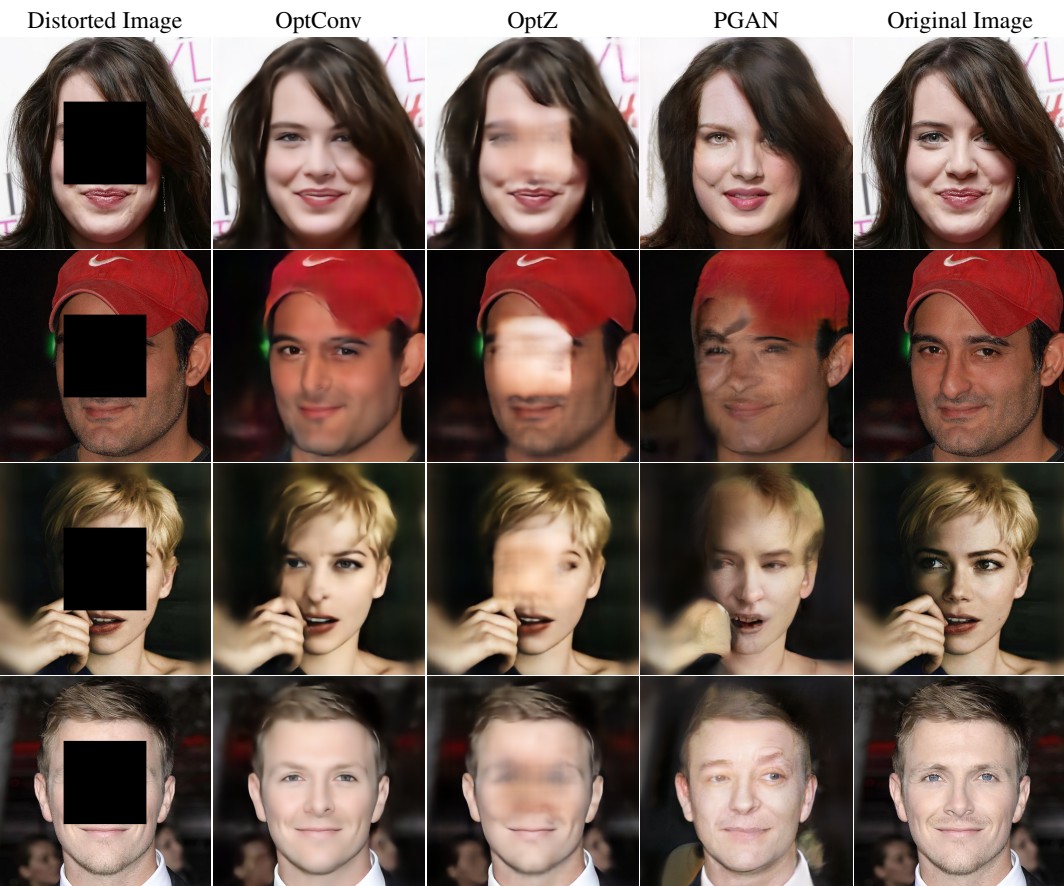

Figure 6: A comparision of optimization over the convolutional manifold (column "OptConv"), the z-space (column "OptZ") and the Progressive GAN (Karras et al., 2018) latent space (column "PGAN") on the CelebA-HQ dataset (Karras et al., 2018).

## 3.2 EXPERIMENTS ON CELEBA-HQ AND THE ROLE OF THE CONVOLUTIONAL MANIFOLD.

For the CelebA-HQ, we have limited comparison of the LCM model to the pretrained progressive GAN model (Karras et al., 2018) published by the authors (this is because proper tuning of the parameters of other baselines would take too much time). On this dataset, LCM uses a latent space of 135k parameters.

Additionally, we use CelebA-HQ to highlight the role of the convolutional manifold structure in the latent space. Recall that the use of the convolutional manifold parameterization is what distinguish the LCM approach from the GLO baseline. The advantage of the new parameterization is highlighted by the experiments described above. One may wonder, if the convolutional manifold constraint is needed at testtime, or if during the restoration process the constraint can be omitted (i.e. if (5) can be used instead of (4) with the generator network $g$ trained with the constraint). Generally, we observed that the use of the constraint at testtime had a minor effect on the CelebA and Bedrooms dataset, but was very pronounced on the CelebA-HQ dataset (where the training set is much smaller and the resolution is much higher).

In Figure 6 and Table 2, we provide qualitative and quantitative comparison between the progressive GAN model (Karras et al., 2018), the LCM model, and the same LCM model applied without the convolutional manifold constraint for the task of inpainting. The full LCM model with the convolutional manifold performed markedly better than the other two approaches. Progressive GAN severely underfit even the known pixels. This is even despite the fact that the training set of (Karras et al., 2018) included the validation set (since their model was trained on full CelebA-HQ dataset). Unconstrained LCM overfit the known pixels while providing implausible inpaintings for the unknown. Full LCM model obtained much better balance between fitting the known pixels and inpainting the unknown pixels.

## 4 CONCLUSION

The results in this work suggest that high-dimensional latent spaces are necessary to get good image reconstructions on desired hold-out sets. Further, it shows that parametrizing these spaces using ConvNets imposes further structure on them that allow us to produce good image restorations from a wide variety of degradations and at relatively high resolutions. More generally, this method can easily be extended to come up with more interesting parametrizations of the latent space, e.g. by interleaving the layers with image-specific and dataset-specific parameters.

The proposed approach has several limitations. First, when trained over very large datasets, the LCM model requires long time to be trained till convergence. For instance, training an LCM on 150k samples of CelebA at $128 \times 128$ resolution takes about 14 GPU-days. Note that the GLO model of the same latent dimensionality takes about 10 GPU-days. On the other hand, the universality of the models means that they only need to be trained once for a certain image type, and can be applied to any degradations after that. The second limitation is that both LCM and GLO model require storing their latent representations in memory, which for large datasets and large latent spaces may pose a problem. Furthermore, we observe that even with the large latent dimensionalities that we use here, the models are not able to fit the training data perfectly suffering from such underfitting. Our model also assumes that the (log)-likelihood corresponding to the degradation process can be modeled and can be differentiated. Experiments suggests that however such modeling needs not be very accurate, e.g. simple quadratic log-likelihood can be used to restore JPEG-degraded images (Appendix H). Finally, our model requires lengthy optimization in latent space, rather than a feedforward pass, at test time. The number of iterations however can be drastically reduced using degradation-specific or universal feed-forward encoders from image-space to the latent space that may provide a reasonable starting point for optimization.

## 5 ACKNOWLEDGEMENTS.

This work has been supported by the Ministry of Science of the Russian Federation (grant 14.756.31.0001). ShahRukh Athar is partially supported by NSF-CNS-1718014, NSF-IIS-1566248 and a gift from Adobe.

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

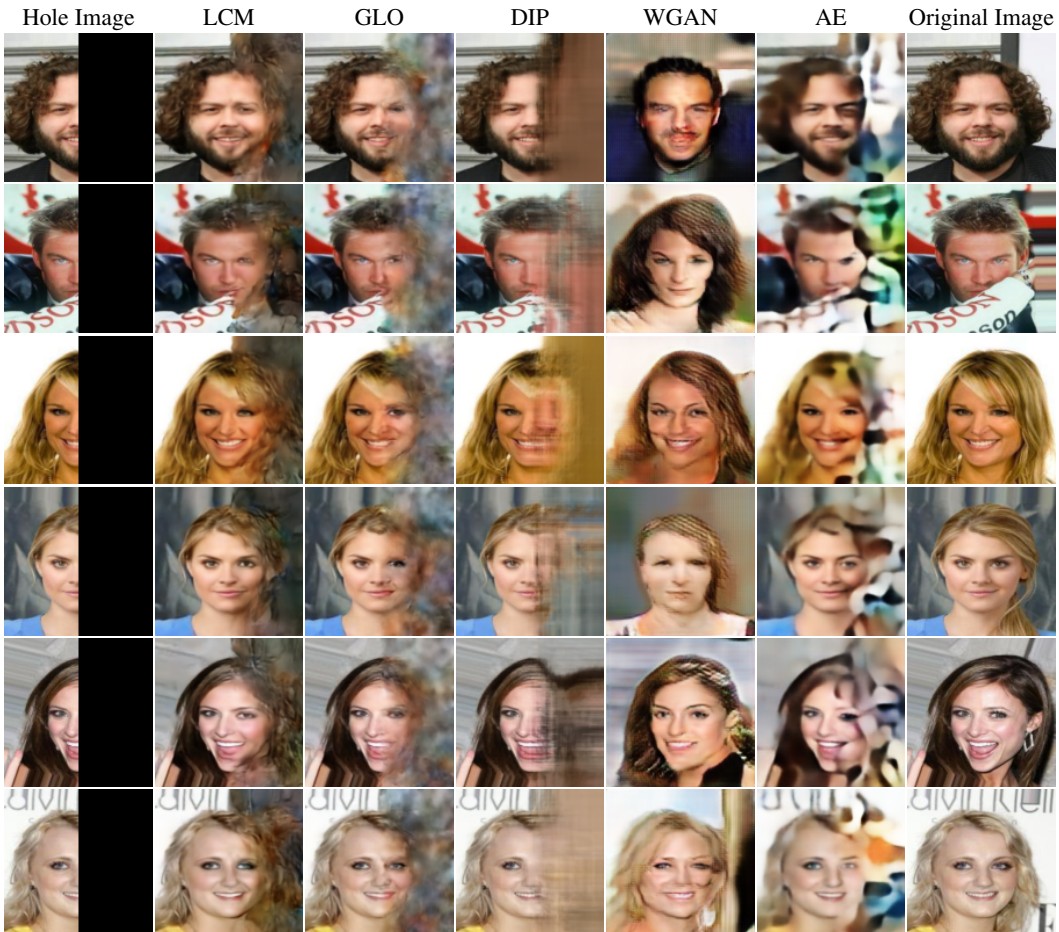

Figure 7: Half-image completion task results on the CelebA dataset ($128 \times 128$ resolution)

## A    OTHER INPAINTING TASKS

Further qualitative comparisons are performed on the CelebA dataset. In Figure 7, we show comparison on the "extreme" task of half-image inpainting. Figure 8 gives a comparison for the task of inpainting where 95% of pixel values are occluded at random. In both cases, the LCM model achieves the best balance of fitting the known evidence and the inpainting quality of known pixels.

## B    VECTORIAL GLO RESULTS

As a baseline in the main text, we have used the variant of the GLO model Bojanowski et al. (2018), where the latent space is organized as maps leading to "fully-convolutional" generator. The latent dimensionality is picked the same as for the the LCM model. Here, we provide evidence that using the original GLO implementation with vectorial-structured latent space, followed by a fully-convolutional layer gives worse results. In particular, we have tried different dimensionality of the latent space (up to 8192, after which we ran out of memory due to the size of the generator). The results for vector-space GLO in comparison with the GLO baseline used in the main text are in Figure 9 and Table 3. The vector based GLO model, despite being trained on latent vector with relatively high dimensionality, clearly underfits.

Distorted Image      LCM      GLO      Original Image

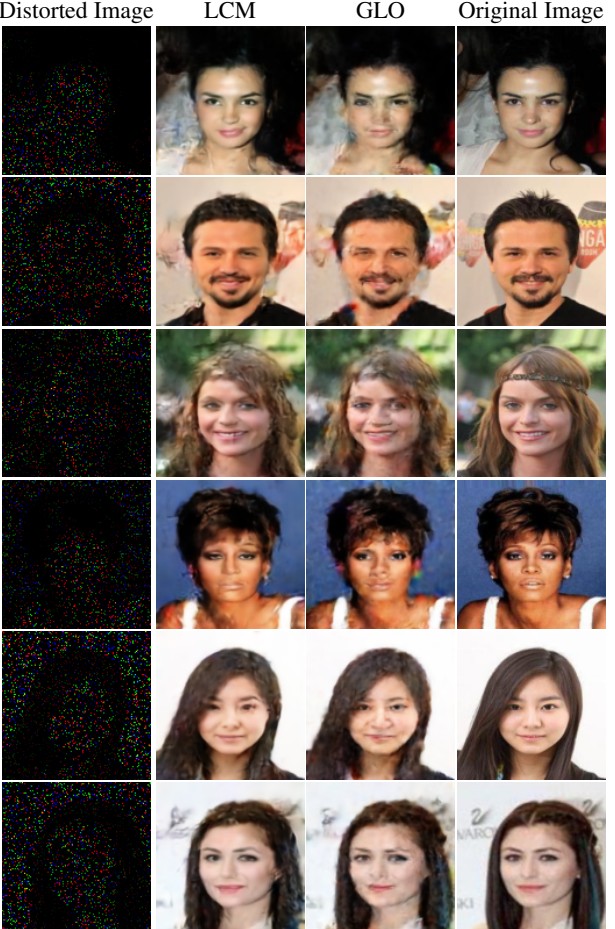

Figure 8: Image inpainting on CelebA with 95% of randomly chosen pixels missing.

Table 3: Training losses of different GLO variants suggesting the underfitting experienced by the vectorial GLO variants.

| Latent Space Type | Latent Space Dimension | Reconstruction Loss (train) |
|---|---|---|
| 3D Map | 24576 | 0.0113 |
| Vector | 8192 | 0.0144 |
| Vector | 4096 | 0.0171 |
| Vector | 2048 | 0.0202 |

Hole Image     24k (Conv)     8192 (vec)     4096 (vec)     2048 (vec)     Original Image

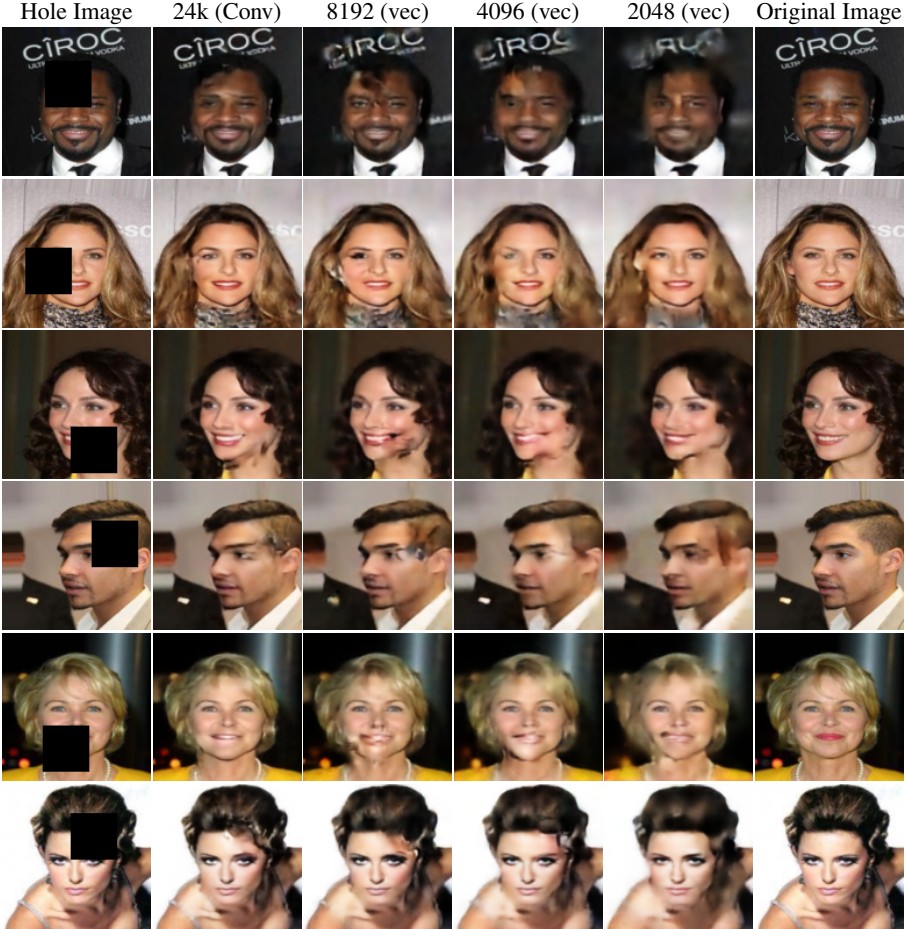

Figure 9: Image inpainting using GLO models with latent spaces of different dimension and structure. The GLO baseline from the main text achieves the best fit to the known pixels and arguably the best inpaintings of the unknown pixels.

Table 4: Reconstruction train losses with different weight penalties using the WGAN-GP.

| L2-penalty Weight | Reconstruction Loss |
|---|---|
| 1 | 0.53 |
| $1e^{-3}$ | 0.3178 |
| $1e^{-5}$ | 0.2176 |
| 0 | 0.1893 |

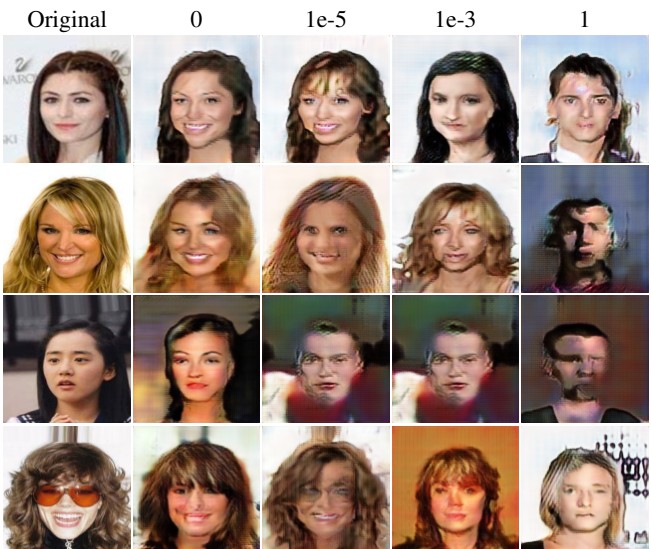

Figure 10: Image reconstruction using the WGAN-GP with gradually increasing penalties on the norm of the latent representation **z** as justified by the probabilistic model behind GANs. Increasing the weight of this penalty (shown above) leads to worse underfitting without improving the quality of the reconstruction. Therefore the comparisons in the main text use the variant without such penalty.

## C  LATENT SPACE PRIOR FOR GANS

Most GAN implementations (including ours) use Gaussian prior when sampling in the latent space. In principle, such prior should be imposed during the restoration process (in the form of an additional term penalizing the squared norm of **z**). We however do not impose such prior in the comparisons in the main text, since it makes the underfitting problem of GANs even worse. In Table 4 we demonstrate that the fitting error for the images from the train set indeed gets worse as the penalty weight is increased. In Figure 10, this effect is demonstrated qualitatively.

## D  ARCHITECTURE DETAILS

The architecture details for the components of the LCM model are as follows:

- Generator Network $g_\theta$: The generator network $g_\theta$ has an hourglass architecture in all three datasets. In CelebA the map size varies as follows: $32 \times 32 \rightarrow 4 \times 4 \rightarrow 128 \times 128$ and the generator has a total of 38M parameters. In Bedrooms the map size varies as: $64 \times 64 \rightarrow 4 \times 4 \rightarrow 256 \times 256$ and the generator has a total of 30M parameters. In CelebAHQ the map size varies as $256 \times 256 \rightarrow 32 \times 32 \rightarrow 1024 \times 1024$ and the generator has a total of 40M parameters. All the generator networks contain two skip connections within them and have a batch-norm and the LeakyReLU non-linearity after every convolution layer.

- Latent Network $f_{\phi_i}$: The latent network used in CelebA128 consists of 4 convolutional layers with no padding. The latent network used in Bedrooms and CelebA-HQ consists of 5 and 7 convolutional layers respectively with no padding.

The code of our implementation is available at the project website.

## E TRAIN/TEST LOSSES

For the sake of completeness we provide losses of LCM and GLO models on the training and test set. We additionally provide the loss if the LCM is optimized over the z-space (i.e the output of $f_\phi$) instead of the parameters of $f_\phi$ (The results shown in row "LCM Z-Space"). In general, the full LCM model has higher loss for train and for test sets, as being more constrained than the other two methods. The additional constraints however allow the LCM model to perform better at image reconstruction tasks.

Table 5: Reconstruction Loss as measured by the L1-Laploss + MSE on the training and test set. LCM has a higher reconstruction error because while LCM and GLO have the same latent space dimensionality, the LCM additionally imposes the convolutional manifold constrain on the latent space which is absent in GLO.

| Model Name | Train Set | Test Set |
|:---:|:---:|:---:|
| **LCM (Ours)** | 0.01306 | 0.01345 |
| **GLO** | 0.01007 | 0.01033 |
| **LCM Z-Space** | 0.01205 | 0.01235 |

## F    MORE RESULTS ON BEDROOMS

In Figure 11, we provide additional inpainting and superresolution results on the Bedrooms dataset for the compared methods.

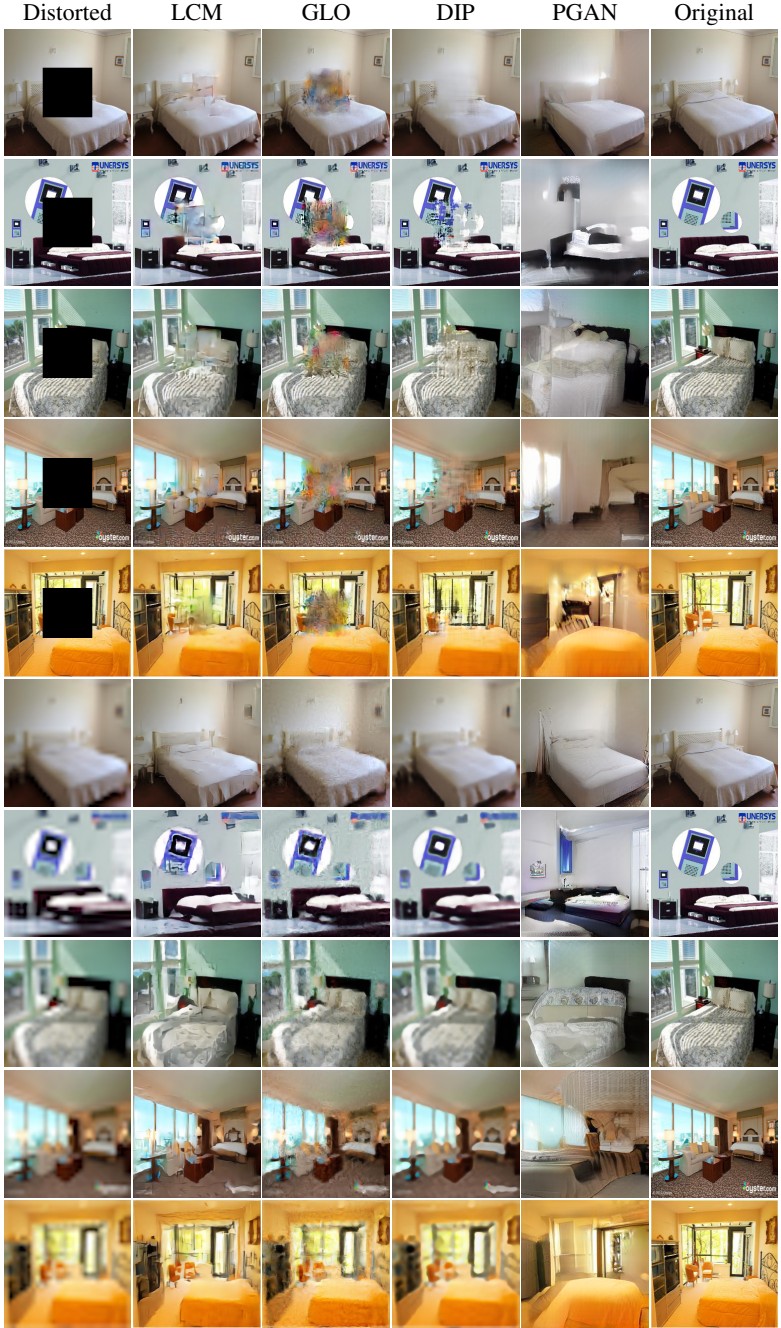

Figure 11: Additional qualitative comparisons on the Bedrooms dataset (see main text for discussion).

## G INTERPOLATIONS IN THE LATENT SPACE

In this section we show the results of performing linear interpolations on the latent space of the convolutional GLO, and LCM and compare it to a linear cross-fade performed in the image space. We start by first finding the best fitting latent parameters (we optimize over $\phi$ for LCM and over $z$ for convolutional GLO) for the source and target images and then perform linear interpolation between them. As can be seen in Figure 12, interpolations in the LCM latent space seem to be smoother and a lot more faithful to the training data distribution than interpolations in convolutional GLO latent space.

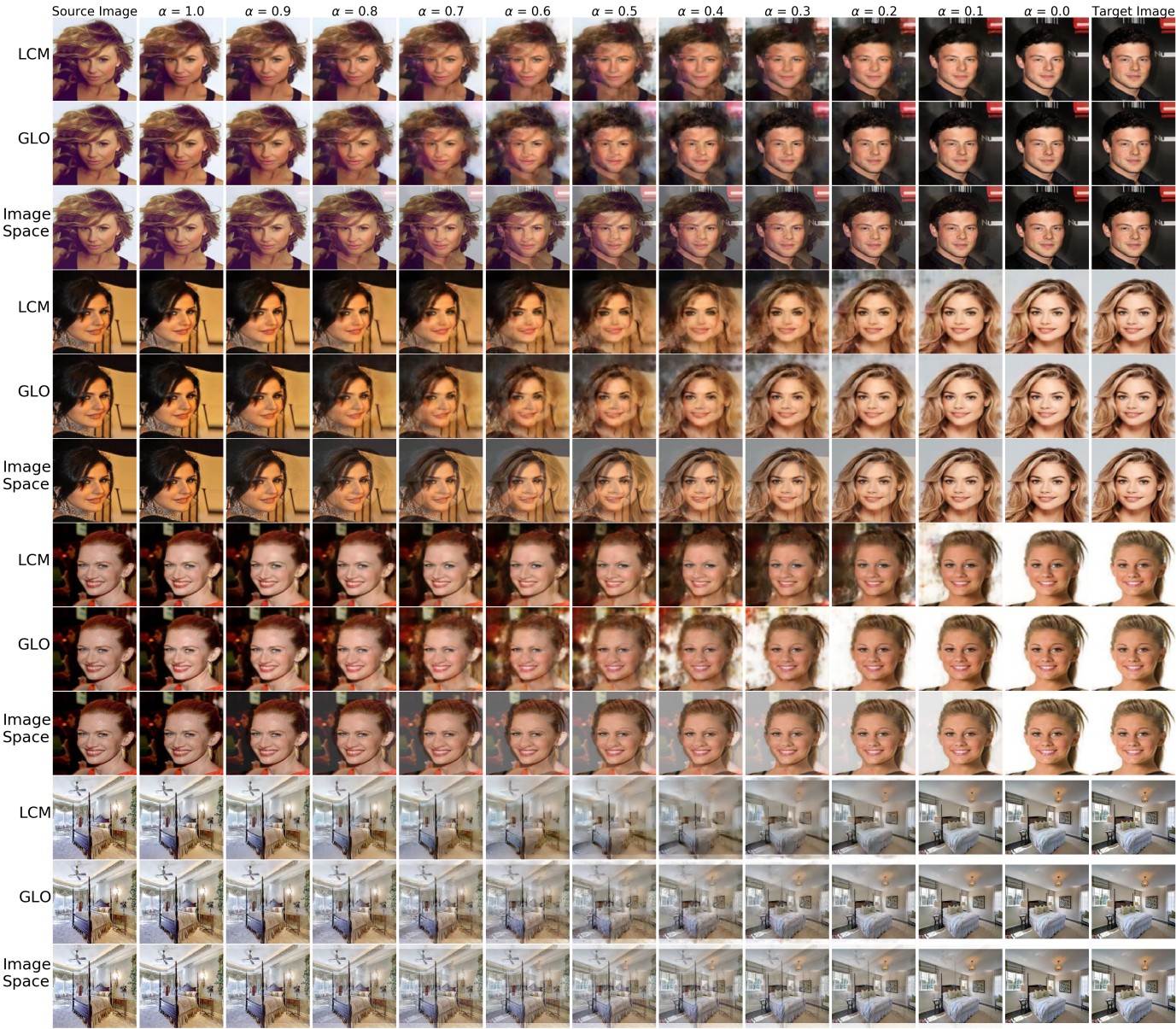

Figure 12: Interpolations in the latent space of the LCM model (top row), the convolutional GLO model (middle row). For the reference, we also provide linear cross-fade in the image pixel space in the bottom row. In the case of our model, the interpolation is performed between $\phi_1$ and $\phi_2$, i.e. along the convolutional manifold. Arguably, LCM interpolations are more plausible, with faces rotating smoothly and with more plausible detailes (e.g. noses) in the case of LCM. Generally, there is noticeably less "double-vision" artefacts. Electronic zoom-in recommended.

## H   JPEG IMAGE RESTORATION

In this section we perform JPEG image restoration using a squared error negative log-likelihood function as a loss. As in the case of inpainting, super-resolution and colorization we perform the optimization over $\phi$ keeping the generator fixed. Results in Figure 13 suggest that LCMs can be used to restore images even when application-specific likelihood function is unknown/hard to model.


Compressed Image       LCM Restoration       Original Image


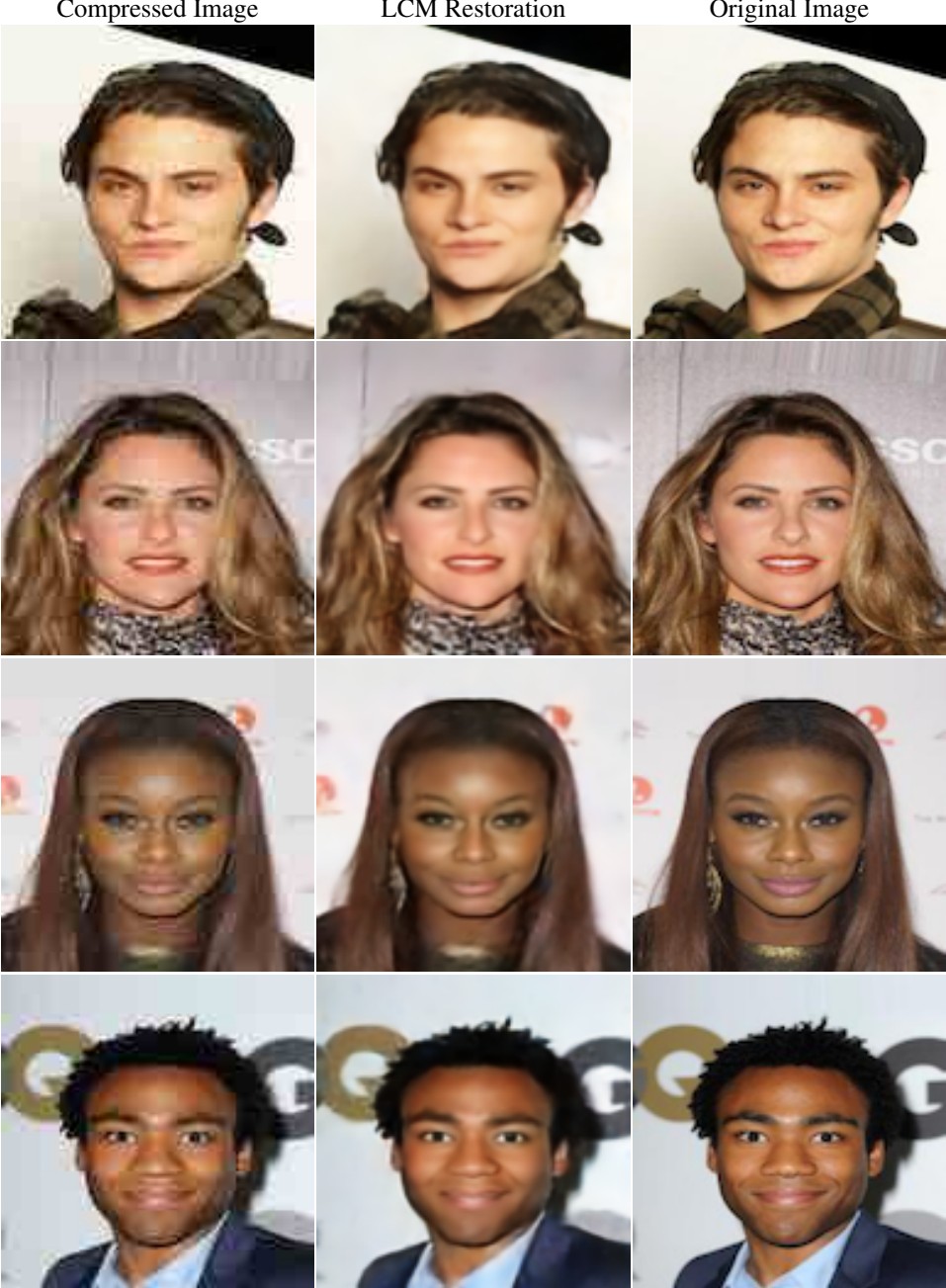

Figure 13: Image restoration from heavy JPEG compression. Left – the input, middle – restored, right – ground truth. Rather than modeling JPEG degradation with a specific likelihood function, we used a simple quadratic (log)-likelihood potential (corresponding to Gaussian noise corruption).

# I UNCONDITIONAL IMAGE GENERATION

In this section, we show the results of unconditional sampling from the LCM latent space. A random subset of $m = 30k$ trained latent ConvNet parameter vectors $\{\phi_1, \ldots, \phi_m\}$ are first mapped to a 512-dimensional space using PCA. We then fit a GMM with 3 components and a full covariance matrix on these 512-dim vectors and sample from it. Figure 14 shows the results of the sampling procedure.

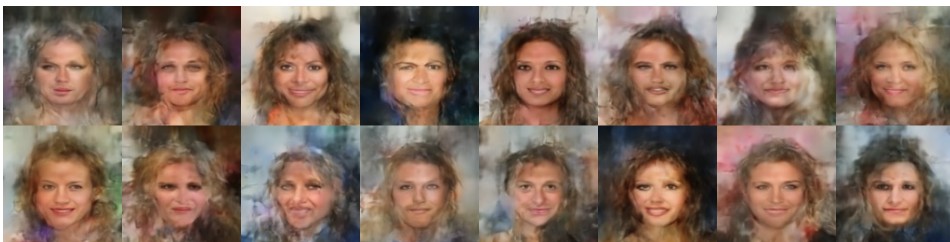

Figure 14: Unconditional Image Generation. We first project the latent parameters, the $\phi$'s, to a lower dimensional space using PCA and then sample from it. The details are given in the text.

