# OpenReview forum: "Latent Convolutional Models"
_ICLR.cc/2019/Conference_

### Official Review · AnonReviewer3 · 2018-11-02
**[Review] Latent Convolutional Models**

**Rating:** 7
**Confidence:** 4

**Review:**

[Summary]
- This work proposes a new complex latent space described by convolutional manifold, and this manifold can map the image in a more robust manner (when some part of the image are to be restored).

[Pros]
- The results show that the latent variable mapped to the image well represents the image, and it will be helpful for the image restoration problem.
- it seems novel to adapt the idea of DIP for defining complex latent space.

[Cons]
- The main concern is that there is no guarantee that the defined latent space is continuous.
It means that it is difficult to judge whether the interpolated point (phi_in, s_in) between two points: (phi_1, s_1) and (\phi_2, s_2), will be matched to the image distribution.
Equation 2 in the paper seems that it just fit the generator parameter theta to map the phi_i and x_i and memorize the mapping between the training images and the given latent convolutional variables.
If the proposed algorithm just memorizes the training image and map them into given the latent convolution, the result cannot justify the proposal that the author proposes a new latent space.

[Summary]
- This work proposes an interesting idea of defining complex latent space, but It is doubtful that this work just memorized the mapping between the training images and the latent convolutional parameters.
- I want to see the (latent space) interpolation test for the proposed latent convolutional space. If the author provides a profound explanation of the problem, I would consider changing the rating.

--------------------------
See the additional comment for the changed rating

---

> ### Author Response · Authors · 2018-11-11
> **Response to R3**
>
> Thank you for the careful review. Fortunately, your main concern though very grave is due to a very simple misunderstanding. We hope that once the misunderstanding is resolved, the rating may be reconsidered.
>
> "Equation 2 in the paper seems that it just fit the generator parameter theta to map the phi_i and x_i and memorize the mapping between the training images and the given latent convolutional variables.
> If the proposed algorithm just memorizes the training image and map them into given the latent convolution, the result cannot justify the proposal that the author proposes a new latent space."
>
> We want to stress that all evaluations and qualitative examples are produced on the _hold-out_ test sets that were not in any way used to train the parameters theta of the generator network. So, we can very confidently say that the reason why the approach works is not memorization of the training set within theta.
>
> "I want to see the (latent space) interpolation test for the proposed latent convolutional space."
>
> We have added latent space interpolations to the appendix G (Figure 12) in the end of the paper. These interpolations were again done on a _hold-out_ set of images. The examples were ``cherry-picked'' for distinctiveness. In more details, in our (biased) view, LCM were always at least as good as other methods, but in some cases, e.g. for pairs of aligned perfectly frontal faces all interpolations look more or less the same, so we picked cases with clear difference between methods. Thank you for suggesting this comparison, it nicely illustrates the effect of the convolutional manifold constraint. If possible, please use zoom-in/large screen to view these results.
>
> "..the interpolated point (phi_in, s_in) between two points: (phi_1, s_1) and (\phi_2, s_2).."
>
> Actually, the s vector is always fixed to some random noise value. I.e. it is not instance specific and is not modified by learning (one can add optimization over s, but in practice this does not change much).

---

> > ### Comment · AnonReviewer3 · 2018-11-23
> > **Response to the Author comment**
> >
> > I agree the comment from the author mostly for my concerns.
> >
> > (1) From the interpolated images, a point in the latent space seems to be matched to corresponding image in the image distribution, which means that it does not simply memorizes the images.
> >
> > (2) By seeing the figure 8, I think this work can be tested in image generation task, either. In final version, I strongly want to see the Pure Image generation result.
> >
> > Based on the comment, I changed my previous rating.

---

### Official Review · AnonReviewer2 · 2018-11-04
**universal image prior with compelling results, but more limited than specialized restoration nets**

**Rating:** 7
**Confidence:** 2

**Review:**

# Summary
The paper proposes to embed natural images in a latent convolutional space of high dimensionality to obtain a universal image prior. Concretely, each image is embedded as a custom parameter vector of a CNN, which turns random noise into the input of a universal generator network to restore the image in pixel space.
Inference for image restoration is performed by minimizing the energy of a likelihood objective while constraining the latent representation of the restored image to be part of the learned latent space. Experiments for inpainting, super-resolution, and colorization are performed to evaluate the proposed method.

# Positive
As mentioned in the paper, I agree that the idea of learning a universal image prior is appealing, since it can be applied to (m)any image restoration tasks without adjustment.
I am not very familiar with the related work, but if I understood correctly, the paper seems to combine deep latent modeling (GLO, Bojanowski et al., 2018) and deep image priors (Ulyanov et al., 2018). The experiments show good results which qualitatively appear better than those of related methods. A user study also shows that people mostly prefer the results of the proposed method.
Did you try other standard restoration tasks, such as image denoising or deblurring? If not, do you think they would work equally well?

# Limitations
While I agree that a universal image prior is valuable, the paper should (briefly) mention what the disadvantages of the proposed approach are:
- A limitation (at least as presented) is that the corruption process has to be known analytically (as a likelihood objective) and must be differentiable for gradient-based inference.
- Furthermore, the disadvantage of the universal prior as presented in the paper is that restoring an image requires optimization (e.g. gradient descent). In contrast, corruption-specific neural nets typically just need a forward pass to restore the image and are thus easier and faster to use.

# Restoration inference
- How dependent is the restoration result with respect to the initialization? For example, when starting gradient descent with the degraded image vs. a random image.
- Roughly, how many iterations and runtime is needed for inference?
- Did you try different optimizers, such as L-BFGS?

---

> ### Author Response · Authors · 2018-11-11
> **Response to R2**
>
> Thank you for the careful review. Here are the responses.
>
> "Did you try other standard restoration tasks, such as image denoising or deblurring? If not, do you think they would work equally well?"
> We have tried denoising (with synthetic noise), where the relative performance is similar. We have not tried deblurring, although we expect the relative performance.
>
> "- A limitation (at least as presented) is that the corruption process has to be known analytically (as a likelihood objective) and must be differentiable for gradient-based inference."
> While technically we do assume that the corruption process is known, it is still possible to apply our approach with simplified (inaccurate) likelihood function. To show that we have added appendix H (Figure 13), which shows how restoration from heavy JPEG artifacts can be done using simple quadratic likelihood functional. The second limitation (need for optimization at test time) is indeed important. We can partially remedy it by adding encoder that would take a corrupted image and output a good starting point in a latent space. We have added discussion/acknowledgement of these limitations to the end of the conclusion section.
>
> "- How dependent is the restoration result with respect to the initialization? For example, when starting gradient descent with the degraded image vs. a random image."
> Our approach cannot start with the degraded image, since we do not know the corresponding latent space initialization. So we always start with a random latent vector. Generally, we found that initializing the latent networks using the same parameters as when the training started worked the best (so we always use the same random vector). Different starting points lead to results with very slightly worse visual quality (the perceptual loss increases by about 0.0006), which are still better than that of competing methods.  Note, that we experimented with different initializations for all the models and chose the one that worked the best for each (to give baselines a fair treatment).
>
> "Roughly, how many iterations and runtime is needed for inference?"
> For a batch of 50 images, it takes about 1000-2000 iterations with takes between 6-12 minutes. Tasks like super-resolution can be done in about 1000 iterations or so and inpainting can take up to 1500-2000 iterations.
>
> "- Did you try different optimizers, such as L-BFGS?"
> Yes, we have tried L-BFGS for inference. We had to use a lower learning rate and were able to produce results similar to that of SGD. Generally, L-BFGS did not offer any significant advantages over SGD.

---

### Official Review · AnonReviewer1 · 2018-11-05
**Latent Convolutional Models**

**Rating:** 6
**Confidence:** 3

**Review:**

This paper proposes to increase the latent space dimensionality  of images, by stacking the latent representation vectors as a tensor. Then convolutional decoder and encoder networks are used to map the original data to latent space and vice versa. The learned latent representations can then be used in a universal framework for multiple tasks such as image inpainting, superresolution and colorization.

The idea of increasing the dimensionality of the latent space, although not sophisticated, seems to be performing very good. Indeed in some of qualitative experiments, the results are surprising. The authors should clarify that how is the training procedure performed in more details. Are test images included in the training the convolutional networks?

---

> ### Author Response · Authors · 2018-11-11
> **Response to R1**
>
> Thank you very much for the review.
> We would like to point out that there are no encoder network in our approach (although one can possibly discuss ways to add it). Also, note that our contribution is not that we only increase the resolution of the latent space, but that we suggest a specific regularization of the latent space (the convolutional manifold) that significantly improves the generalizability of the resulting latent model.
>
> "Are test images included in the training the convolutional networks?"
> All results (qualitative, quantitative, user study) are performed on hold-out sets, that were not used to train the parameters of the decoder (i.e. theta). The only exception is the progressive GAN baseline, for which there is a mix of training and test sets (since for the comparison we just reuse author-provided models trained on complete sets). This gives an advantage to the pGAN baseline (admittedly not a very big one, since GANs struggle to fit the training sets). To reiterate, all results of OUR method (LCM) are computed strictly on the hold-out test sets.
>
> To train our model we use the Laplacian-L1 along with an MSE term with a weight of 1.0. We noticed that the MSE term speeds up convergence without affecting the results by much. The optimization is carried out using stochastic gradient descent with a learning rate of 1.0. We note that the code for the paper and the experiments will be released for reproducibility.

---

### Meta-Review · Area_Chair1 · 2018-12-14
**Latent model for images - presents convincing results on image impainting+**

**Confidence:** 4
**Recommendation:** Accept (Poster)

**Metareview:**

The reviewers are in general impressed by the results and like the idea but they also express some uncertainty about how the proposed actually is set up. The authors have made a good attempt to address the reviewers' concerns.